# The Role of Olfactomedin 2 in the Adipose Tissue–Liver Axis and Its Implication in Obesity-Associated Nonalcoholic Fatty Liver Disease

**DOI:** 10.3390/ijms24065221

**Published:** 2023-03-09

**Authors:** Andrea Barrientos-Riosalido, Laia Bertran, Mercè Vilaró-Blay, Carmen Aguilar, Salomé Martínez, Marta Paris, Fàtima Sabench, David Riesco, Jessica Binetti, Daniel Del Castillo, Cristóbal Richart, Teresa Auguet

**Affiliations:** 1Grup de Recerca GEMMAIR (AGAUR)—Medicina Aplicada (URV), Departament de Medicina i Cirurgia, Universitat Rovira i Virgili (URV), Institut d’Investigació Sanitària Pere Virgili (IISPV), 43007 Tarragona, Spain; 2Servei Anatomia Patològica, Hospital Universitari Joan XXIII Tarragona, Mallafré Guasch, 4, 43007 Tarragona, Spain; 3Servei de Cirurgia, Hospital Sant Joan de Reus. Departament de Medicina i Cirurgia, URV, IISPV, Avinguda Doctor Josep Laporte, 2, 43204 Reus, Spain; 4Servei Medicina Interna, Hospital Universitari de Tarragona Joan XXIII, Mallafré Guasch, 4, 43007 Tarragona, Spain

**Keywords:** olfactomedin 2, nonalcoholic fatty liver disease, adipose tissue, lipid metabolism

## Abstract

This study’s objective was to assess the involvement of olfactomedin 2 (OLFM2), a secreted glycoprotein related to lipid metabolism regulation, in nonalcoholic fatty liver disease (NAFLD) mediated by the adipose-tissue–liver axis. OLFM2 mRNA expression was analyzed in subcutaneous (SAT) and visceral (VAT) adipose tissue by RT–qPCR. The cohort included women with normal weight (n = 16) or morbid obesity (MO, n = 60) who were subclassified into normal liver (n = 20), simple steatosis (n = 21), and nonalcoholic steatohepatitis (NASH, n = 19) groups. The results showed that OLFM2 expression in SAT was enhanced in MO individuals and in the presence of NAFLD. Specifically, OLFM2 expression in SAT was increased in mild and moderate degrees of steatosis in comparison to the absence of it. Moreover, OLFM2 expression in SAT was negatively correlated with interleukin-6 levels. On the other hand, OLFM2 expression in VAT decreased in the presence of NASH and exhibited a positive correlation with adiponectin levels. In conclusion, OLFM2 in SAT seems to be implicated in hepatic lipid accumulation. Additionally, since we previously suggested the possible implication of hepatic OLFM2 in NAFLD progression, now we propose a possible interaction between the liver and SAT, reinforcing the potential implication of this tissue in NAFLD development.

## 1. Introduction

In recent years, nonalcoholic fatty liver disease (NAFLD) has grown to be a major public health problem, and its incidence has increased drastically, with a prevalence of approximately 25% in the global adult population [1]. NAFLD is a liver disorder characterized by the abnormal accumulation of intrahepatic fat in the absence of other etiologies. It is considered a multisystemic disease and a hepatic manifestation of metabolic syndrome (MetS). NAFLD is associated with type 2 diabetes mellitus (T2DM), obesity, and insulin resistance (IR) [2]. Its natural history begins with simple steatosis (SS), which occurs when there is more than 5% fat in hepatocytes without evidence of hepatic dysfunction. However, this condition can develop into nonalcoholic steatohepatitis (NASH), a more serious disorder characterized by severe steatosis, hepatocellular ballooning, and lobular and/or portal inflammation with or without hepatic fibrosis. This severe condition can evolve into cirrhosis or hepatocellular carcinoma if it is not treated [2]. The pathogenesis of NAFLD is described by the multiple hit hypothesis, which postulates that there are many factors simultaneously impacting the liver that cause the development of NAFLD [3]. In addition, crosstalk with other organs, such as adipose tissue, can also affect the liver when the tissue is dysfunctional, as in obese or diabetic patients, resulting in a flow of free fatty acids (FFAs) and inflammatory adipokines to the liver and inducing hepatic fatty accumulation, liver inflammation, and hepatic damage (Figure 1).

In recent years, the importance of the interaction between adipose tissue and the liver in the progression of NAFLD has been noted [4]. There are two types of white adipose tissue: subcutaneous adipose tissue (SAT), which is located under the skin and exhibits increased expression of proinflammatory genes in patients with morbid obesity (MO) [5,6]; and visceral adipose tissue (VAT), which covers internal organs and sends an increased flow of adipokines and FFAs to the liver through the portal circulation [5,6,7]. Conditions of excessive fat accumulation in VAT have been shown to be associated with the development of metabolic disorders and a predisposition for NAFLD [8,9]. In addition, epidemiological research has shown that SAT expansion induces insulin sensitivity and lowers the incidence of T2DM, while VAT build-up increases metabolic risk and total mortality [5]. In addition, the pathogenesis of NAFLD involves many metabolic pathways that have not yet been fully elucidated. For this reason and because this disease has a very invasive gold-standard diagnostic method, although it is very accurate, and the lack of specific pharmacological treatments accepted by regulatory agencies [10,11,12], more studies are necessary on the molecular mechanisms involved in NAFLD to find new therapeutic targets.

Olfactomedin 2 (OLFM2) is a secreted glycoprotein [13] related to the regulation of lipid metabolism, insulin resistance, and obesity [14] that has been implicated in different diseases, such as ocular glaucoma [15,16] and hepatocellular carcinoma [17]. The Human Protein Atlas (HPA) reported that OLFM2 was mostly expressed in the brain, but high expression was also found in the liver and adipose tissue [18]. Therefore, in our previous study, we evaluated the potential role of hepatic OLFM2 in obesity-associated NAFLD. We conducted a study using liver biopsies, and we found increasing relative OLFM2 mRNA expression as hepatic disease became more severe, suggesting a potential key role of OLFM2 in the progression of NAFLD [19]. However, this protein has not been well studied, especially in terms of liver or metabolic diseases, making it difficult to draw conclusions.

Since we first described the association between OLFM2 and NAFLD and increased expression of OLFM2 in adipose tissue was reported in the HPA [18], we wanted to investigate the relative mRNA expression of OLFM2 in the current research in SAT and VAT and study its role in adipose-tissue–liver-axis-mediated NAFLD in a well-established cohort of women with normal weight (NW) or MO who had different degrees of NAFLD involvement.

## 2. Results

### 2.1. Patients’ Baseline Characteristics

The clinical and biochemical values of the patients are shown in Table 1. The cohort included women who were separated into two groups based on their body mass index (BMI): those who were normal weight (NW, BMI < 25 kg/m^2^, n = 16) and those than showed MO (BMI ≥ 40 kg/m^2^, n = 60). Subsequently, MO patients were subdivided into three groups based on liver histology: normal liver (NL, n = 20), SS (n = 21), and NASH (n = 19). The cohort members were comparable in terms of age, diastolic blood pressure (DBP), systolic blood pressure (SBP), and low-density lipoprotein cholesterol (LDL-C). Apart from exhibiting higher weight and BMI in NL and SS groups than the NW group, we found enhanced values of the homeostatic model assessment method for insulin resistance (HOMA1-IR), glucose, insulin, alanine aminotransferase (ALT), and gamma-glutamyltransferase (GGT) in the SS and NASH groups compared to the NW cohort. In addition, we observed an increase in waist, glycosylated hemoglobin (HbA1c), and triglyceride (TG) levels in the MO group, which was the opposite of what was observed with high-density lipoprotein cholesterol (HDL-C), which was increased in the NW group. There were lower levels of cholesterol in the SS group compared to the NW group and an increase in aspartate aminotransferase (AST) levels in NASH subjects. Compared with NL patients, we reported an increase in glucose and alkaline phosphatase (ALP) levels in SS and NASH patients, while in the SS group, ALP was increased compared to that in the NASH group. In addition, we analyzed the liver histology of the cohort and some proinflammatory cytokines in women with MO, such as interleukin (IL)-8 and (IL)-6; and some anti-inflammatory mediators, such as adiponectin and IL-10. However, these cytokines did not exhibit significant differences between the groups.

### 2.2. Evaluation of the Relative mRNA Expression of OLFM2 in the Adipose Tissue of Patients with NW or MO

First, we determined the relative mRNA expression of OLFM2 in VAT and SAT in the presence of obesity (BMI < 25 kg/m^2^ or BMI ≥ 40 kg/m^2^). We observed only a significant increase in OLFM2 mRNA expression in the SAT of MO subjects, as shown in Figure 2A. However, in VAT (Figure 2B), we did not find significant differences between the groups.

### 2.3. Evaluation of the Relative mRNA Expression of OLFM2 in Adipose Tissue Based on Hepatic Histology

Since we had previously reported that hepatic OLFM2 could play a role in the progression of NAFLD and higher expression of OLFM2 mRNA was observed in the SAT of MO subjects, we analyzed the mRNA expression of OLFM2 according to liver involvement. First, we divided the patients according to whether they had NAFLD. In this regard, we observed an enhancement in relative OLFM2 mRNA abundance in the SAT of NAFLD patients compared to subjects with healthy livers (Figure 3A). However, we did not find significant differences in VAT (Figure 3B). Second, we evaluated the relative mRNA expression of OLFM2 in adipose tissues based on the different histopathological grades of NAFLD (NL, SS, and NASH). SAT exhibited a significant increase in OLFM2 mRNA expression in the SS and NASH groups compared to the NL and NW control groups, as shown in Figure 3C. However, in VAT, we observed nonsignificant differences between the groups (Figure 3D).

### 2.4. Evaluation of Relative OLFM2 mRNA Expression in Adipose Tissues Based on Liver Steatosis Degree

Then, to further examine the connection between OLFM2 mRNA expression in adipose tissue and NAFLD, we analyzed the relative mRNA expression of OLFM2 in adipose tissues according to the different degrees of liver steatosis (absent, mild, moderate, and severe). In SAT, we observed enhanced expression of OLFM2 in mild and moderate steatosis stages in comparison with the absence of liver steatosis, as shown in Figure 4A. However, we did not find significant differences in the expression of OLFM2 depending on the different degrees of steatosis in VAT (Figure 4B).

### 2.5. Evaluation of Relative OLFM2 mRNA Expression in Adipose Tissue in the Presence of NASH and in Relation to NASH-Related Parameters

Subsequently, we further examined the relationship between the expression of OLFM2 in adipose tissue and NASH in depth, and we analyzed the relative abundance of OLFM2 in the adipose tissue samples that was classified according to the presence or absence of NASH. Meanwhile, in SAT, we reported nonsignificant differences (Figure 5A), but in VAT, we reported lower OLFM2 mRNA expression in NASH patients than in non-NASH patients (Figure 5B). Given this significant difference, we evaluated the expression of OLFM2 in relation to different parameters related to NASH, such as portal or lobular inflammation and hepatocyte ballooning. Regarding lobular inflammation, we did not find significant differences in SAT (Figure 5C), while in VAT, we observed lower OLFM2 mRNA in patients with lobular inflammation than in subjects without lobular inflammation (Figure 5D). Regarding portal inflammation, in SAT, we did not find significant differences between the groups (Figure 5E), while in VAT, we reported lower expression of OLFM2 in the presence of portal inflammation than in its absence, as shown in Figure 5F. However, when analyzing the relative mRNA abundance of OLFM2 in terms of the presence of ballooning, we reported nonsignificant differences in adipose tissue among the groups.

### 2.6. Correlation between Relative OLFM2 mRNA Expression in SAT and VAT and Clinical and Biochemical NAFLD-Related Parameters

To test our previous hypothesis, we analyzed correlations between OLFM2 in adipose tissue and different clinical and biochemical parameters related to NAFLD. We observed a negative association between the relative abundance of OLFM2 mRNA in SAT and circulating levels of IL-6, as shown in Figure 6A. On the other hand, in VAT, we found a positive association between the mRNA expression of OLFM2 and adiponectin levels (Figure 6B).

## 3. Discussion

The most interesting findings of our study lie in the fact that we propose a possible interaction between OLFM2 in adipose tissue and NAFLD pathogenesis. Since we previously reported an association between hepatic OLFM2 expression and NAFLD progression [19], and given that this protein, which is highly expressed in adipose tissue [18], is associated with the regulation of lipid metabolism [14,16], we believe that it is both interesting and novel to study the expression of OLFM2 in adipose tissue (SAT and VAT) and its implication in obesity-associated NAFLD [4]. Our main findings were that the relative mRNA expression of OLFM2 in SAT was increased in patients with MO compared to NW subjects. Furthermore, OLFM2 expression was enhanced in the presence of NAFLD. Moreover, OLFM2 expression in SAT was increased in mild and moderate degrees of steatosis in comparison to the absence of it. Regarding OLFM2 in VAT, we reported a decrease in expression in the presence of NASH. Finally, we reported a negative association between OLFM2 mRNA expression in SAT and circulating IL-6 levels. Additionally, we found a positive correlation between OLFM2 expression in VAT and adiponectin levels.

Our first finding shows that, in SAT, the relative mRNA expression of OLFM2 was increased in patients with MO compared to NW subjects. However, in VAT, we did not find significant differences between the groups. Similarly, Gonzalez-Garcia et al. carried out a study comparing OLFM2-null mice and wild-type mice and reported that OLFM2-null mice exhibited lower body weights [14]. In addition, mice with OLFM2 expression in the hypothalamus exhibited the opposite phenotype and had increased weight. This research showed that OLFM2 was involved in the regulation of energy metabolism [14]. Therefore, our results reinforce the findings of Gonzalez-Garcia et al., since our MO patients exhibited increased expression of OLFM2 in SAT, suggesting a potential role of OLFM2 in the regulation of energy metabolism in this tissue. In this sense, some studies have described the involvement of SAT in human metabolism, as well as an important role in lipotoxicity or insulin resistance [20,21]. Unfortunately, we did not find increased OLFM2 expression in VAT, which has been reported to be related to metabolic disruption [22]. Additionally, the expression of various genes in SAT and VAT in obesity was previously shown to be different [5,23]. This phenomenon could also occur in NAFLD and could explain our results, suggesting a potential role of OLFM2 in SAT in lipid metabolism or in inflammatory processes [24].

We reported increased expression of OLFM2 mRNA in the SAT of patients with NAFLD compared to patients without NAFLD. When the cohort was classified according to the degree of NAFLD, we observed an increase in OLFM2 mRNA abundance in patients with SS and NASH compared to controls. These results were similar to our previous study in which hepatic OLFM2 mRNA expression was increased as hepatic involvement became more severe [19]. Some studies reported that SAT exhibits increased expression of proinflammatory genes in patients with MO [5,6]; similarly, Plessis et al. demonstrated the significance of the SAT gene set in both the early stage of NAFLD and its progression to NASH [25], which may support the hypothesis that OLFM2 in SAT could play a role in NAFLD progression in the same way that OLFM2 affects the liver. There have been few reports on OLFM2, and it is difficult to understand the specific mechanisms and their implication in NAFLD.

Because the expression of OLFM2 in adipose tissue mirrors its expression in the liver and increases as liver conditions worsen, and given the regulatory role that OLFM2 may play in lipid metabolism [14], we decided to evaluate the relative mRNA expression of OLFM2 in terms of the degree of steatosis in NAFLD patients. We found that OLFM2 expression was higher in subjects with mild and moderate steatosis than in those without it. This finding is consistent with the previous results and reinforces our hypothesis that OLFM2 may be related to the development of NAFLD and contribute to the regulation of lipid metabolism, as mentioned in Gonzalez-Garcia et al. [14]. However, additional studies are needed to confirm the effect of OLFM2 in adipose tissue on NAFLD.

Concerning VAT, we reported a decrease in OLFM2 mRNA expression in NASH and in the presence of portal and lobular inflammation compared to the absence of these factors. It has been postulated that VAT plays a fundamental role in the development of NAFLD and metabolic diseases [26] due to its connection through the portal vein, directly exposing the liver to the flow of FFAs and proinflammatory factors [27]. This result may seem contradictory, since, in a previous study, we reported an increase in the expression of hepatic OLFM2 mRNA in patients with NASH and in subjects with lobular inflammation [19]. However, OLFM2 expression had not previously been evaluated in VAT. In addition, since most of our results related OLFM2 in SAT to NAFLD, we concluded that the key role of OLFM2 in VAT in inflammation related to NAFLD could be different or even contrary to that in the liver [20,21], as well as in mitochondrial respiration, which is a decreased in VAT but not in SAT in obese individuals with NAFLD [28].

We examined the correlations between the expression of OLFM2 and different biochemical and clinical variables that are related to NAFLD. We found a negative association between OLFM2 mRNA expression in SAT and circulating IL-6 levels. In this way, it is important to note that IL-6 is a proinflammatory adipokine that is usually increased in NAFLD [29]. However, our patients did not exhibit significant differences in IL-6 levels between the groups, probably due to the low-grade chronic inflammation that is caused by MO, which alters inflammatory mediator levels [30,31], as well as SAT metabolism [8]. Although obesity is closely related to NAFLD, excess storage of visceral fat is considered equally or more important and can mask the NAFLD condition; therefore, visceral-fat reduction is necessary to promote metabolic changes in NAFLD [32].

Additionally, we found a positive correlation between OLFM2 mRNA expression in VAT and circulating adiponectin levels. In this regard, adiponectin is an anti-inflammatory cytokine that is typically decreased in NAFLD [33]. Thus, this correlation was consistent with the association with OLFM2 mRNA expression in VAT, and we found decreased expression of this molecule in VAT in NASH, which would decrease adiponectin levels; however, in our subjects, we did not find significant differences in adiponectin, probably due to the abovementioned low-grade chronic inflammation [34].

Thanks to this study, we were able to evaluate the potential role of OLFM2 in adipose tissue in obesity-associated NAFLD. Meanwhile, we studied a cohort composed of only women, and subjects with liver biopsy exhibited MO. We found a potential role of OLFM2 in SAT in NAFLD, but these results are preliminary and cannot be extrapolated to other populations. Therefore, further research is needed in this field.

## 4. Materials and Methods

### 4.1. Patients

The Institut Investigació Sanitària Pere Virgili (IISPV, Tarragona, Spain) approved this study through the institutional review committee (CEIm; 23c/2015; 11 May 2015). The cohort consisted of 76 Caucasian women with NW (BMI > 25 kg/m^2^, n = 16) and MO (BMI ≥ 40 kg/m^2^, n = 60). Written informed consent was obtained from all participants; SAT, VAT, and, during planned laparoscopic bariatric surgery, liver biopsies were collected. These were the conditions for exclusion: (1) an ethanol intake greater than 10 g/day or other toxins; (2) menopausal women or women taking contraceptives to prevent interference from hormones that can bias glucose and lipid metabolism, as well as cytokine determinations; (3) patients with infectious disease, neoplastic disease, or acute or chronic liver disease other than NAFLD; and (4) patients receiving fibrates because this medication can interfere with the metabolism of some metabolites derived from the microbiota studied in this work.

### 4.2. Sample Size

To realize our objective, sample size was calculated using a GRANMO calculator (IMIM, Barcelona, Spain). Accepting an alpha risk of 0.05 and a beta risk of less than 0.2 in a two-sided test, a minimum of 22 controls and 67 cases are needed to recognize as statistically significant an odds ratio greater than or equal to 0.13. A proportion of the exposed subjects in the control group was estimated to be 0.25%, using the POISSON test.

### 4.3. Liver Pathology

An experienced hematopathologist used the method of Kleiner and Brunt [35,36] to stratify the liver samples, using hematoxylin and eosin and Masson’s trichrome stains (see Figure 7 for visual information of the histology). Women with MO were classified according to their hepatic histopathology into NL (n = 20) and NAFLD (n = 40). Subjects with NAFLD were subclassified into SS (micro/macrovesicular steatosis without inflammation or fibrosis, n = 21) and NASH (Brunt grades 1–2, n = 19). In our group, there was no liver fibrosis porto-portal in any of the NASH patients.

### 4.4. Biochemical Analyses

Prior to bariatric surgery and having fasted the night before, specialized nurses collected blood samples into tubes with or without ethylenediaminetetraacetic acid through a BD Vacutainer^®^ system. These samples were then separated into aliquots of serum and plasma centrifugation (3500 rpm, 4 °C, and 15 min). A conventional automatic analyzer was used to analyze the biochemical parameters, and the IR was estimated through the homeostatic model for IR (HOMA1-IR). Cytokines, such as interleukin IL-6, IL-8, IL-10, TNF-α, and adiponectin, were determined in all samples of the cohort by multiplex sandwich immunoassays and MILLIPLEX MAP Human Adipokine Magnetic Bead Panel 1 (HADK1MAG-61K, Millipore, Billerica, MA, USA), the MILLIPLEX MAP High Sensitivity Human T-Cell Panel (HSTCMAG28SK, Millipore, Billerica, MA, USA), and the Bio-Plex 200 instrument, according to manufacturer’s instructions. All of these analyses were evaluated at the Omic Sciences Center (Eurecat, Reus, Spain), and the physical, anthropometric, and biochemical evaluations were performed on the entire cohort.

### 4.5. Gene Expression in Liver

The hepatic and adipose tissue samples were collected during bariatric and conserved in tubes with RNAlater (Qiagen, Hilden, Germany) at 4 °C. Then samples were processed and stored at −80 °C. RNeasy mini kit (Qiagen, Barcelona, Spain) was used to extract total RNA from adipose tissue. Reverse transcription to cDNA was performed with the High-Capacity RNA-to-cDNA Kit (Applied Biosystems, Madrid, Spain). Real time quantitative polymerase chain reaction (PCR) was carried out with the TaqMan Assay predesigned by Applied Biosystems for the detection of OLMF2 (Hs01017934_m1). The expression of each gene was calculated and standardized to the expression of glyceraldehyde-3-phosphate dehydrogenase (GAPDH) (Hs02786624_g1); after, they were normalized using the control group (NW) as a reference. All reactions were duplicated in 96-well plates, using the QuantStudio™ 7 Pro Real-Time PCR System (Applied Biosystem, Foster City, CA, USA). Determination of OLFM2 mRNA expression was analyzed in all adipose tissue samples obtained from the cohort (n = 76 (NW, n = 16; NL, n = 20; SS, n = 21; NASH, n = 19)); the raw data of the mRNA relative expression of OLFM2 in adipose tissue (CT/CQ values) are found in the Appendix A Appendix A.

### 4.6. Statistical Analysis

The data were analyzed using the SPSS/PC+ for Windows statistical package (version 27.0; SPSS, Chicago, IL, USA). The distribution of variables was obtained using the Kolmogorov–Smirnov test, and the different comparative analyses were assessed using Mann–Whitney U test to compare groups. Using Spearmen’s method, the coefficient of correlation (rho) between variables was calculated. All results were expressed as the median and the interquartile range (25th–75th). The *p*-values < 0.05 were statistically significant. Graphics were elaborated using GraphPad Prism software (version 7.0; GraphPad, San Diego, CA, USA).

## 5. Conclusions

In this study, we found that OLFM2 in SAT could regulate lipid metabolism involved in the progression of NAFLD. Since we previously suggested a possible implication of hepatic OLFM2 in NAFLD progression, we now propose a possible interaction between the OLFM2 of SAT and the liver, reinforcing the fact that a SAT–liver axis may be implicated in NAFLD development.

## Figures and Tables

**Figure 1 ijms-24-05221-f001:**
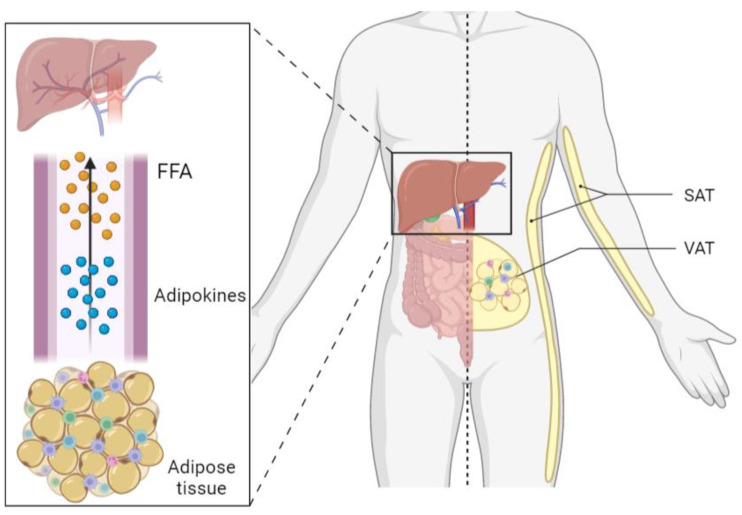
Crosstalk between adipose tissue and liver. Dysfunctional adipose tissue sends FFA and adipokines through the circulation to the liver, being implicated in nonalcoholic fatty liver disease. SAT, subcutaneous adipose tissue; FFA, free fatty acids; VAT, visceral adipose tissue.

**Figure 2 ijms-24-05221-f002:**
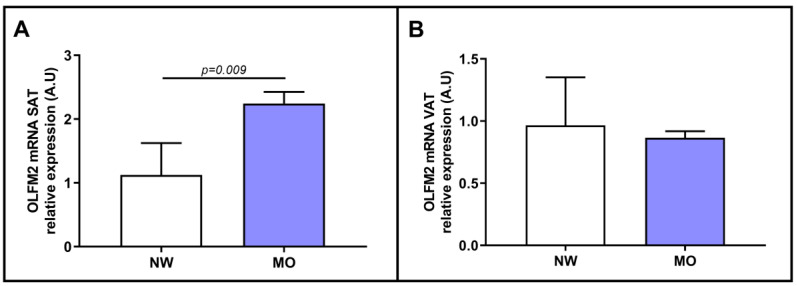
Comparison of mRNA abundance of OLFM2 in SAT (**A**) and in VAT (**B**) between patients with NW and MO. NW, normal weight; MO, morbid obesity; OLFM2, olfactomedin 2; VAT, visceral adipose tissue; SAT, subcutaneous adipose tissue; A.U, arbitrary units. Mann–Whitney test was used to calculate differences between groups, and *p* < 0.05 was considered statistically significant.

**Figure 3 ijms-24-05221-f003:**
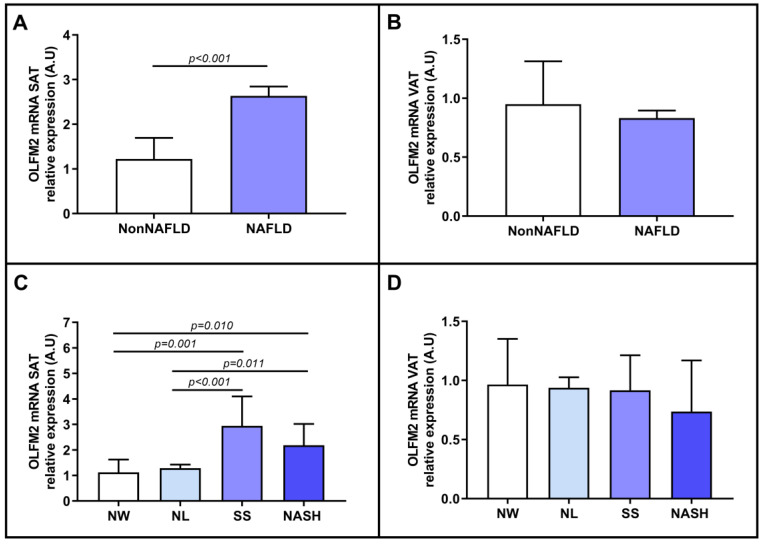
Differential relative mRNA expression of OLFM2 in SAT (**A**) and VAT (**B**) between patients classified according to the presence or the absence of NAFLD, and being classified by NW, NL, SS, and NASH in SAT (**C**) and VAT (**D**). OLFM2, olfactomedin 2; VAT, visceral adipose tissue; SAT, subcutaneous adipose tissue; NAFLD, nonalcoholic fatty liver disease; NASH, nonalcoholic steatohepatitis; NW, normal weight; MO, morbid obesity; SS, simple steatosis; A.U arbitrary units. Mann–Whitney test was used to calculate differences between groups, and *p* < 0.05 was considered statistically significant.

**Figure 4 ijms-24-05221-f004:**
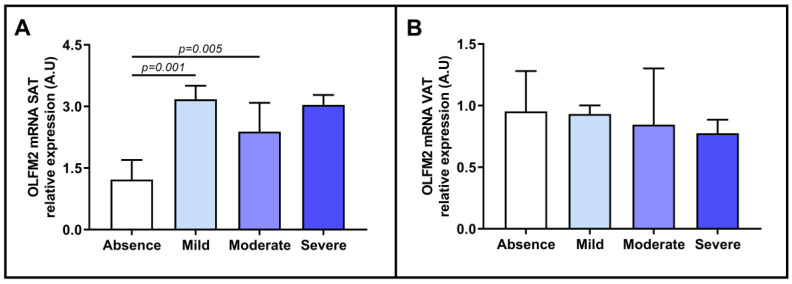
Comparison of mRNA abundance of OLFM2 in SAT (**A**) and in VAT (**B**) samples between subjects classified into absence, mild, moderate, and severe, according to different grades of steatosis. OLFM2, olfactomedin 2; VAT, visceral adipose tissue; SAT, subcutaneous adipose tissue; A.U arbitrary units. Mann–Whitney test was used to calculate differences between groups, and *p* < 0.05 was considered statistically significant.

**Figure 5 ijms-24-05221-f005:**
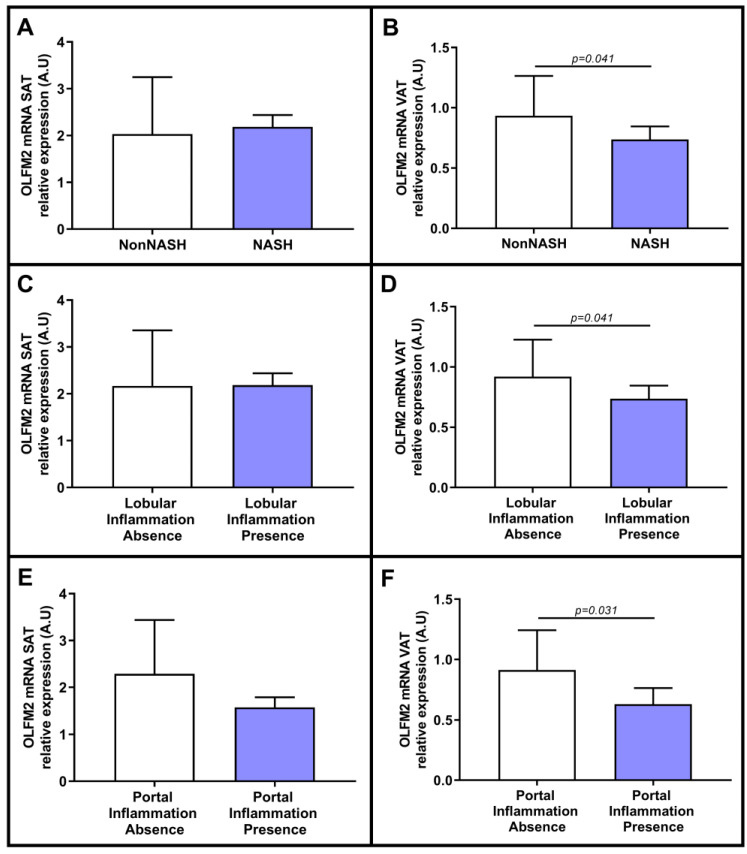
Comparison of mRNA expression of OLFM2 in SAT (**A**) and VAT (**B**) between absence or presence of NASH. Differential relative mRNA abundance of OLFM2 in SAT (**C**) and VAT (**D**) between absence or presence of lobular inflammation and portal inflammation absence or presence (**E**) in SAT and (**F**) in VAT. OLFM2, olfactomedin 2; VAT, visceral adipose tissue; SAT, subcutaneous adipose tissue; NASH, nonalcoholic steatohepatitis; A.U, arbitrary units. Mann–Whitney test was used to calculate differences between groups, and *p* < 0.05 was considered statistically significant.

**Figure 6 ijms-24-05221-f006:**
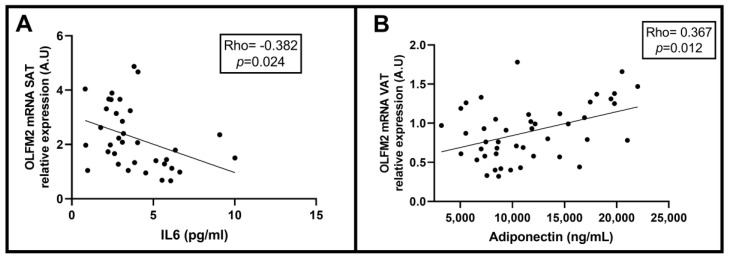
(**A**) Significant negative correlations between OLFM2 mRNA expression in SAT and circulating levels of IL-6. (**B**) Significant positive associations between OLFM2 mRNA expression in VAT and adiponectin levels. OLFM, olfactomedin; IL, interleukin; SAT, subcutaneous adipose tissue; VAT, visceral adipose tissue; A.U, arbitrary units. The Spearman test was used to calculate the correlation coefficient (rho), and *p* < 0.05 was considered statistically significant.

**Figure 7 ijms-24-05221-f007:**
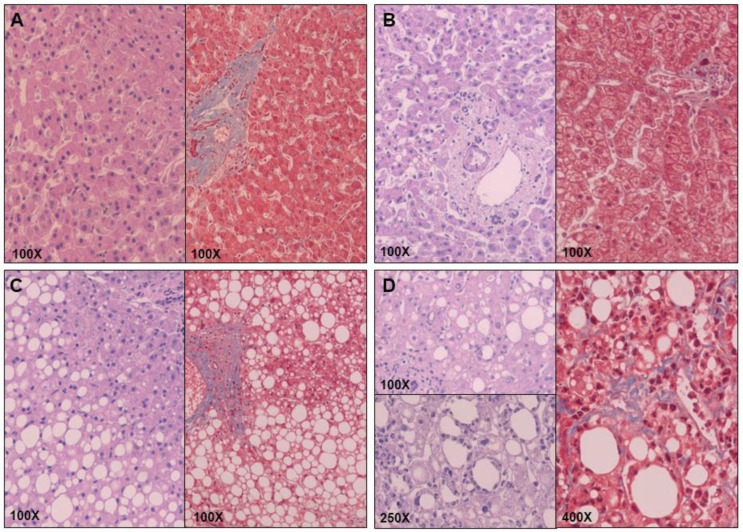
Histological staining with hematoxylin and eosin (purple) and Masson’s trichrome (red) of women with (**A**) NW and NL, (**B**) MO and NL, (**C**) MO and SS, and (**D**) MO and NASH.

**Table 1 ijms-24-05221-t001:** Biochemical and anthropometric variables of patients of the study.

		MO (n = 60)		
Variables	NW (n = 16)	NL (n = 20)	SS (n = 21)	NASH (n = 19)
Weight (kg)	57.75 (52.10–62.00)	119,00 (108.50–134.00) *	115.40 (111.63–129.80) *	110.50 (104.00–121.68)
Age (years)	42.93 (37.06–51.93)	42.30(35.00–52.04)	46.01 (36.11–53.1615)	48.41 (40.62–57.88)
Waist (cm)	72.00 (70.00–84.00)	124.20 (119.75–130.00) *	125.00 (116.75–134.00) *	123.50 (115.00–126.75) *
BMI (kg/m^2^)	22.47 (21.29–24.19)	43.62 (41.56–48.83) *	44.94 (42.07–46.83) *	44.54 (40.95–47.23)
DBP (mmHg)	70.00 (66.00–75.00)	63.00 (58.00–71.00)	62.00 (59.00–73.25)	68.00 (60.25–78.00)
SBP (mmHg)	117.00 (110.00–125.00)	120.00 (100.00–132.00)	116.00 (108.00–127.00)	117.00 (105.00–132.00)
HOMA1-IR	1.30 (0.85–2.17)	2.07 (1.22–3.45)	2.44 (1.27–3.05) *	1.90 (1.45–6.16) *
Glucose (mg/dL)	83.58 (64.81–94.34)	85.59 (77.58–92.84)	93.09 (88.08–106.11) * ^$^	93.09 (88.09–106.11) * ^$^
Insulin (mUI/L)	7.00 (4.90–9.62)	9.70 (5.59–16.21)	9.80 (6.94–14.10) *	9.54 (5.68–26.02) *
HbA1c (%)	5.10 (4.70–5.40)	5.60 (5.30–5.75) *	5.55 (5.3–5.85) *	5.55 (5.30–5.85) *
TG (mg/dL)	86.00 (57.25–110.25)	106.00 (93.00–136.00) *	117.50 (84.00–165.50) *	130.50 (99.25–187.50) *
Cholesterol (mg/dL)	183.25 (163.53–209.50)	170.00 (150.15–214.50)	165.70 (138.75–189.50) *	162.00 (150.50–213.25)
HDL-C (mg/dL)	62.40 (48.45–73.00)	40.20 (31.50–48.50) *	43.50 (33.25–46.75) *	38.00 (34.50–44.00) *
LDL-C (mg/dL)	112.20 (89.90–130.00)	108.80 (95.20–141.80)	103.10 (77.20–124.86)	93.40 (79.30–126.83)
AST (UI/L)	20.00 (16.00–26.00)	19.50 (15.00–36.25)	21.00 (17.00–31.00)	30.00 (18.00–43.50) *
ALT (UI/L)	17.00 (12.50–25.00)	21.00 (16.00–37.00)	29.50 (22.00–35.00) *	33.50 (18.75–41.00) *
GGT (UI/L)	14.00 (10.00–31.00)	17.00 (13.00–23.00)	21.00 (16.25–32.75) *	26.00 (19.75–34.00) *
ALP (Ul/L)	65.00 (51.50–88.00)	57.50 (47.75–71.75)	73.50 (62.00–86.00) ^$^	61.00 (53.25–74.50) ^$ #^
**Liver Histology:**				
Steatosis Grade 0/1/2/3	-	20/0/0/0	0/15/6/0	0/3/16/0
Inflammation Grade 0/1/2/3	-	20/0/0/0	21/0/0/0	2/18/5/0
Ballooning 0/1/2	-	20/0/0/0	21/0/0/0	2/17/0/0
Fibrosis Stage 0/1/2/3/4	-	-	-	-

MO, morbid obesity; NW, normal weight; NL, normal liver; SS, simple steatosis; NASH, nonalcoholic steatohepatitis; BMI, body mass index; DBP, diastolic blood pressure; SBP, systolic blood pressure; HOMA1-IR, homeostatic model assessment method of insulin resistance; HbA1c, glycosylated hemoglobin; TG, triglycerides; HDL-C, high density lipoprotein cholesterol; LDL-C, low density lipoprotein cholesterol; AST, aspartate aminotransferase; ALT, alanine aminotransferase; GGT, gamma-glutamyltransferase; ALP, alkaline phosphatase; IL, interleukin. Data are expressed as the median (interquartile range). * Significant differences between the NW group and the other groups (*p* < 0.05). $ Significant differences between the NL cohort and the other groups (*p* < 0.05). # Significant differences between the SS patients and the other women (*p* < 0.05).

## Data Availability

Data is unavailable due to privacy.

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
