# Peer review of "The Role of Olfactomedin 2 in the Adipose Tissue–Liver Axis and Its Implication in Obesity-Associated Nonalcoholic Fatty Liver Disease"

_ijms, 2023, doi:10.3390/ijms24065221_

Round 1

Reviewer 1 Report

The relationship between NAFLD progression and adipose tissue is investigated through the lens of OLF2M.   Appears publishable with minor revisions.

I commend the inclusion of the NW group, which is often overlooked.  Surprisingly some studies even omit a NL group, so it is good to see that here as well. Several clinicals do not show expected changes.  ALT and AST are expected to change more particularly with NASH.  Strange that IR does not increase, as it is a well known reporter on NAFLD.  Do the authors have any insights on what might have happened there? The NL size is small, but it probably does not prevent publication.

Table 1 needs age included.  Also, if waist is available it would be desirable to have waist in Table 1. 

Table 1 needs to have the histology summarized (typically summarized as ballooning, lob inflammation, steatosis grading, fibrosis staging).  For an example see the characteristics table in Ioannou et al

 https://www.mdpi.com/2218-1989/10/4/168

(reviewer has no relationship to that paper, just offering it as an example)

Comments on intro: it is more accurate to say that NAFLD is regarded as hepatic aspect of MetS.   Also NASH can be reversed.   Should be noted that progression of NASH strongly related to insulin resistance  (IR) as well.  Should explain ‘difficult to diagnose’ since a liver biopsy is invasive but is also very accurate.  Fibroscan, MRI, etc. are useful too.

Honestly not sure what to do about it, but the correlations in Fig 6 while apparently significant, have tremendous scatter and would hesitate to read too much into it.

I'm not sure that the non-NASH vs NASH comparisons are meaningful.  Is Non-NASH a combination of NL and SS? 

Miscellaneous:

Style: Don’t t know the journal policy, as some journals even prohibit saying novel/first, but reviewer suggests to avoid ‘novelty’.

Style: try to find more specific wording than ‘may play a role’ .

Style: Avoid saying ‘to support our hypothesis’ but instead ‘to test our hypothesis’

One author lists a Hotmail email account which is a little unusual.

Reviewer 2 Report

The manuscript reports the development of olfactomedin 2 (OLFM2), a secreted glycoprotein related to lipid metabolism regulation, in NAFLD mediated by the adipose tissue-liver axis. It proposed a possible interaction between the liver and SAT, reinforcing the potential implication of this tissue in NAFLD development.

Comments on the manuscript are outlined as follows:

1.      The design of the work is not obvious, there is no obvious rationale about this research. At the same time, the binding between this research and previous similar research about OLFM2 is weak.

2.      The manuscript should be revised for some typo errors. For example, "BMI > 25 kg/m2"on line 98 should be changed to " BMI < 25 kg/m2".

3.      There is a decrease in cholesterol levels in the SS group compared to the NW group. This is unnormal. Please recheck your raw data.

The relative abundance of OLFM2 mRNA in SAT is negative related to circulating levels of IL-6. At the same time, the patients do not exhibit significant differences in IL-6 levels between NW and MO groups. Therefore, it is wondered that how OLFM2 regulates lipid metabolism though IL-6.

Reviewer 3 Report

In this study, the authors attempted to determine the crosstalk of olfactomedin 2 in the adipose tissue-liver axis during obesity-associated NAFLD. The below listed concerns need to be satisfactorily addressed by the authors to arrive a decision.

1.       It is observed that the IEC approval number (CEIm; 23c/2015; 11 May 2015) of this study is same as of their previous manuscript - Bertran et al. (2022). However, the cohort, sample size and liver pathology between these two studies are different. This must be clarified.

2.       The values in Table-1 for the weight, TG between SS and NASH groups looks similar to that of their previous study - Bertran et al. (2022). Therefore, authors must submit the raw data for table-1.

3.       The authors must include the results of haematoxylin/eosin & Masson’s trichrome staining of liver sections.

4.       How many samples in each group were subjected to mRNA expression studies? There is no mention of this in methodology. The CT/CQ values must be included in the results.

5.       Why the authors did not determine the protein expression of OLFM2 by western blot method?

Round 2

Reviewer 2 Report

The authors did not address my previous concerns.

Reviewer 3 Report

The authors have satisfactorily addressed the questions and therefore revised manuscript is recommended for publication. 

Author Response

We thank you for reviewing our manuscript and for being satisfied with the responses and changes made